# Moist and warm conditions in Eurasia during the last glacial of the Middle Pleistocene Transition

María Fernanda Sánchez Goñi [1,2] ✉, Thomas Extier [2], Josué M. Polanco-Martínez [3,4], Coralie Zorzi [2], Teresa Rodrigues[5] & André Bahr [6]

The end of the Middle Pleistocene Transition (MPT, ~ 800-670 thousand years before present, ka) was characterised by the emergence of large glacial ice-sheets associated with anomalously warm North Atlantic sea surface temperatures enhancing moisture production. Still, the direction and intensity of moisture transport across Eurasia towards potential ice-sheets is poorly constrained. To reconstruct late MPT moisture production and dispersal, we combine records of upper ocean temperature and pollen-based Mediterranean forest cover, a tracer of westerlies and precipitation, from a subtropical drill-core collected off South-West Iberia, with records of East Asia summer monsoon (EASM) strength and West Pacific surface temperatures, and model simulations. Here we show that south-western European winter precipitation and EASM strength reached high levels during the Marine Isotope Stage 18 glacial. This anomalous situation was caused by nearly-continuous moisture supply from both oceans and its transport to higher latitudes through the westerlies, likely fuelling the accelerated expansion of northern hemisphere ice-sheets during the late MPT.

The strong non-linear behaviour of the climate system in response to the Milankovitch astronomical forcing is particularly well exemplified during the Middle Pleistocene Transition (MPT, ~1.2–0.67 Ma) when 100,000-year ice age cycles emerged without any significant change in orbital cyclicities dominated by the combination of 23,000 and 41,000-year cycles[1]. Imbrie et al.'s conceptual model[2] shows that the growth and decay of ice-sheets amplify, through albedo, winds and ocean mass properties, the system's initial modest responses to orbital forcing by changing the transport of heat to boreal latitudes and atmospheric $CO_2$ concentration. At the end of the MPT (Marine Isotope Stage-MIS 19 to 17, ~800–670 ka), which is marked by the manifestation of large continental ice-sheets, paradoxical responses are observed with warm sea surface temperatures (SST) in the North

Atlantic mid-latitudes generating excess moisture that fuelled ice-sheet growth in Europe and further north[3–5]. This climatic evolution culminated in the maximum southern extent of the Eurasian inland ice-sheets during MIS 16 at ~650 ka[6] contemporaneously with a shift to greater ice accumulation in North America[7]. Within the end of the MPT, MIS 18 (centred at ~730 ka) is a warm glacial in the North Atlantic Ocean[3,8] and represents an anomaly during this period. However, the cause of the mild expression of MIS 18 remains unclear yet. Its glacial limits are unknown and geomorphological data suggests that the glaciation was limited to the high latitudes, especially in the Southern Hemisphere[9]. Furthermore, little is known about the Eurasian environments and atmospheric climate during the final stages of the MPT. We hypothesise that the combination of unusual warming in the mid/

[1]Ecole Pratique des Hautes Etudes (EPHE, PSL University), Paris, France. [2]Univ. Bordeaux, CNRS, Bordeaux INP, EPOC, UMR 5805, 33600 Pessac, France. [3]Unit of Excellence GECOS, IME, University of Salamanca, 37007 Salamanca, Spain. [4]Basque Centre for Climate Change (BC3), 48940 Leioa, Spain. [5]Divisão de Geologia e Georecursos Marinhos, Instituto Português do Mar e da Atmosfera, Rua Alfredo Magalhães Ramalho, 6, 1495-006 Lisboa, Portugal. [6]Institute of Earth Sciences, Heidelberg University, Im Neuenheimer Feld, 234, 69120 Heidelberg, Germany. ✉e-mail: maria.sanchez-goni@u-bordeaux.fr

high-latitude North Atlantic, strong SST gradient in the tropical West North Pacific Ocean and intermediate glacial background conditions should have enhanced long-term moisture production leading to interglacial climates in the subtropical latitudes of Eurasia during MIS 18, and thus ice accumulation in the southern sectors of Eurasia.

Here, we document the Eurasian environments and climates during the final part of the MPT using high-resolution pollen-based vegetation data, a proxy for precipitation, and alkenone-based SST from IODP Site U1385 located in the South-West Iberian margin (37°34.285'N, 10°7.562'W, 2578 m depth) combined with published sea subsurface temperature records from the same site[3] (Fig. 1). These records are compared with geochemical[10] and grain size[11,12] loess records accounting for the evolution of the East Asian monsoon in North-East China, and alkenone-based SST for the tropical and sub-polar West North Pacific Ocean[13,14] (Fig. 1). Robust statistical analysis techniques applied to the paleoclimatic records combined with iLO-VECLIM model simulations were used to identify the main atmospheric and oceanographic processes underlying the warm ocean-maximum ice growth paradox at the end of the MPT (Methods).

## Results and discussion
### Interglacial climate in South-West Europe during MIS 18 glacial
We qualitatively estimate forest cover changes using arboreal versus non-arboreal pollen percentages. This simple approach that acknowledges the non-linear relationship between plant abundances and pollen percentages[15], is supported by a wide array of studies on the modern relationship between pollen assemblages and vegetation (e.g., refs. 16,17) and, specifically, by the recent work of the pollen representation of the vegetation in the Tagus basin (South-West Iberia)[18], the most important source of pollen preserved at IODP Site U1385. We infer the strong development of the Mediterranean forest cover, mainly composed of deciduous *Quercus* (~40–60 pollen %) and sclerophyllous trees and shrubs (evergreen *Quercus, Olea, Phillyrea, Pistacia* and *Cistus*, ~10 pollen %), during the MIS 18d-b interval, 740 to 725 ka, with a brief forest setback at around 733 ka (MIS 18c) (Fig. 2, Supplementary Fig. S1). This interval is bracketed by the dominance of semi-desert landscapes and heathlands indicating cold-dry and cold-humid climates during MIS 18e and MIS 18a, respectively. The forest pollen percentage reached 60 % at c. 730 ka, a similar value characterising

other interglacials in South-West Iberia such as MIS 13 (~500 ka[19]), the last interglacial (MIS 5, ~130 ka[20]) and the Holocene[21]. Moreover, our data record all the phases characterising the succession of vegetation during interglacial periods in the Mediterranean region (*Juniperus* and *Betula* pioneer woodlands/deciduous and evergreen *Quercus-Olea-Phillyrea-Pistacia* forest/deciduous and evergreen *Quercus* woodlands) (Fig. S1)[22]. Site U1385 indicates that the expansion of the deciduous oaks parallels the development of the Mediterranean sclerophyllous taxa, with a synchronous maximum expansion. The pollen assemblages recorded at this site during the warm phases are very similar to the modern ones inferred from samples collected in the deciduous oak woods of the Tagus basin characterised by ~10–20% of sclerophyllous pollen taxa from the thermomediterranean belt and ~40–60% of deciduous oak pollen taxa from the mesomediterranean belt[18]. No arboreal trees extinct at present in Europe are recorded during this interval at Site U1385. The vegetation inferred from the pollen assemblages at Site U1385 is, therefore, very close to the present-day Mediterranean forest of South-West Iberia, dominated by broadleaf trees, such as the oak and mixed sclerophyll forests[23].

Based on transient model simulations with time-varying insolation and atmospheric $CO_2$ concentrations[21], the strong Mediterranean forest cover development reflects high amount of winter precipitation, and thus relatively zonal and weak westerlies directed towards southern Europe. This interpretation is supported by the high statistical correlation between the NAO index, controlling the intensity and direction of the westerlies, i.e., winter precipitation, and the present-day Mediterranean forest cover changes[24]. The moderate values (~10%) of sclerophyllous plants (evergreen *Quercus, Olea, Pistacia, Phillyrea* and *Cistus*) reflect furthermore a weak seasonal climate characterised by relatively cold and wet summers, compared to other interglacials and, particularly, to MIS 5, marked by sclerophyllous plant values peaking at 20%. Furthermore, the record of a few pollen grains of the summer-drought intolerant tree *Castanea*[25] at c. 725 ka highlights the temperate and humid climate during MIS 18, with the westerlies affecting southern Iberia all over the year. SST at the South-West Iberian margin also reached the highest values, ~18 °C, (Fig. 2) and no substantial freshwater fluxes affected this margin during this interval as indicated by the $\%C_{37:4}$ record (Fig. 2). Upper ocean warming off South-West Iberia during MIS 18 was caused by enhanced subsidence

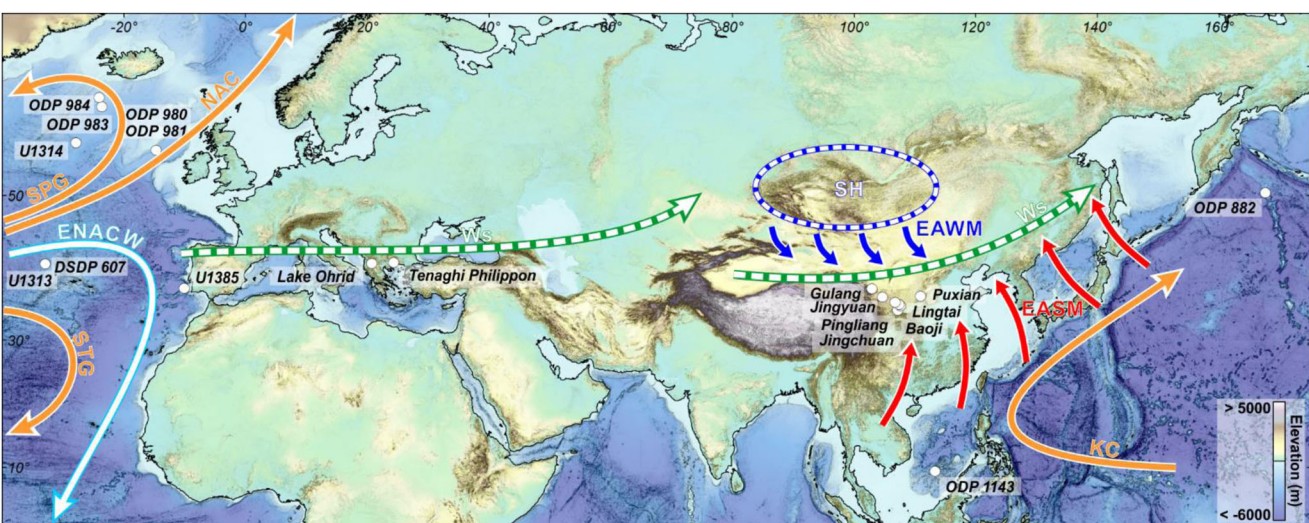

**Fig. 1 | Location of the sites discussed in the text.** IODP Site U1385 (this study), ODP Sites 980, 981, 983, 984, U1313, U1314 and DSDP 607 (ref. 4), Lake Ohrid[40], Tenaghi Philippon[39], Chinese Loess Plateau selected sequences: Gulang[10] and Pingliang, Jingshuan, Baoji, Lingtai, Puxian[11,12], Jingyuan[41], ODP Site 882 (ref. 14), ODP Site 1143 (ref. 13). NAC North Atlantic Current, STG Subtropical Gyre, SPG: Subpolar Gyre, ENACW: Eastern North Atlantic Central Waters, KC Kuroshio Current, EASM East Asia summer monsoon, EAWM East Asia winter monsoon, SH Siberian High, after[78]. Green arrow: present-day westerlies during a low index of the North Atlantic Oscillation (centred at ~40°–45°N[49]), reaching more southern latitudes during the glacial periods as shown by data and model simulations for the Last Glacial Maximum[79].

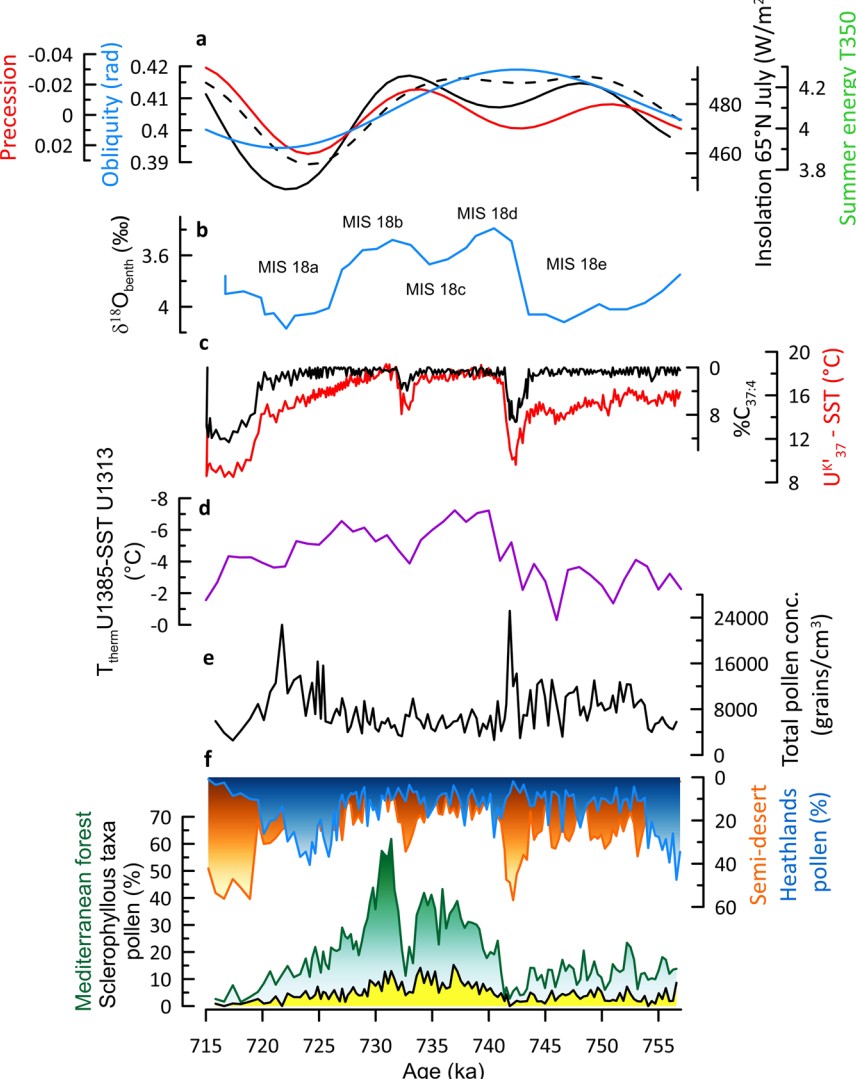

**Fig. 2 | Paleoclimatic records of Marine Isotope Stage (MIS) 18 from the Iberian margin. a** insolation at 65°N in July (black), obliquity (blue) and precession (red)[26], summer energy (dashed black line), T350 defines the number of summer days in which daily insolation is above 350 W/m² (ref. 27), **b** δ[18]O of benthic foraminifera from IODP Site U1385 (blue)[30], sub-stages of MIS 18 follow[80], **c** alkenone-based sea surface temperature (SST, red) and percentages of C[37:4], tracer of freshwater pulses in the Iberian margin, from IODP Site U1385 (black, this study), **d** temperature gradient between T thermocline from IODP Site U1385 (37°N) and SST from IODP Site U1313 (41°N)[3], **e** total pollen concentrations (this study), **f** semi-desert (orange) and heathlands (Ericaceae, blue) pollen percentages, Mediterranean forest (green) pollen percentages mainly composed of deciduous oaks and sclerophyllous (evergreen *Quercus, Pistacia, Olea, Phillyrea* and *Cistus*, yellow) (this study).

of warm East North Atlantic Central Waters (ENACW) resulting in the generation of relatively warm glacial thermocline waters off Iberia[3] (Fig. 2). The resultant extensive upper ocean heat reservoir provided a substantial source of heat and moisture at that time off Iberia that likely favoured the strong development of the regional forest cover during the MIS 18 glacial.

### A warm and wet increasing trend in South-West Europe from MIS 19 to MIS 17

The entire period from MIS 19 to MIS 17 was characterised by an enhanced subduction of warm mid-latitude surface waters to the thermocline off the Iberian margin giving rise to sustained high SST at Site U1385 (ref. 3, Fig. 3). The mid-latitude sourcing of thermocline waters at Site U1385 is illustrated by the gradient between thermocline temperature ($T_{therm}$) at Site U1385 and SST at mid-latitude Site U1313 (41°N) (thereafter "$T_{therm}$ SST gradient"), with a low gradient indicating enhanced subduction of warm mid-latitude waters in contrast to a high gradient indicative of a subpolar source[3]. Gradients of ~6 °C during

warm phases of MIS 19-17 are among the lowest for the entire interval MIS 44-14 (ref. 3) and provide evidence for the high production of warm thermocline waters off Iberia. Atmospheric moisture provided by the accumulation of warm subtropical waters in the mid-latitude eastern North Atlantic could have contributed through the westerlies to the recorded extended mild-humid and forested conditions in central and South-East Europe from MIS 21 to 17[3,5]. Surprisingly, the subduction of warm waters to thermocline level during MIS 18 was similar and even stronger compared to both interglacials MIS 17 and 19, respectively, despite that insolation[26], summer energy[27], atmospheric $CO_2$ concentrations[28] and relative sea level[29,30] were lower during MIS 18 (Fig. 3).

The comparison of the glacial MIS 18 forest expansion with those during the previous and succeeding interglacials is striking (Fig. 3). We observe that the strongest Mediterranean forest development occurred during MIS 17, centred at 700 ka, reflecting a high amount of regional winter precipitation[31]. In contrast, MIS 19 (~785 ka), under the influence of both similar ice volume and higher atmospheric $CO_2$

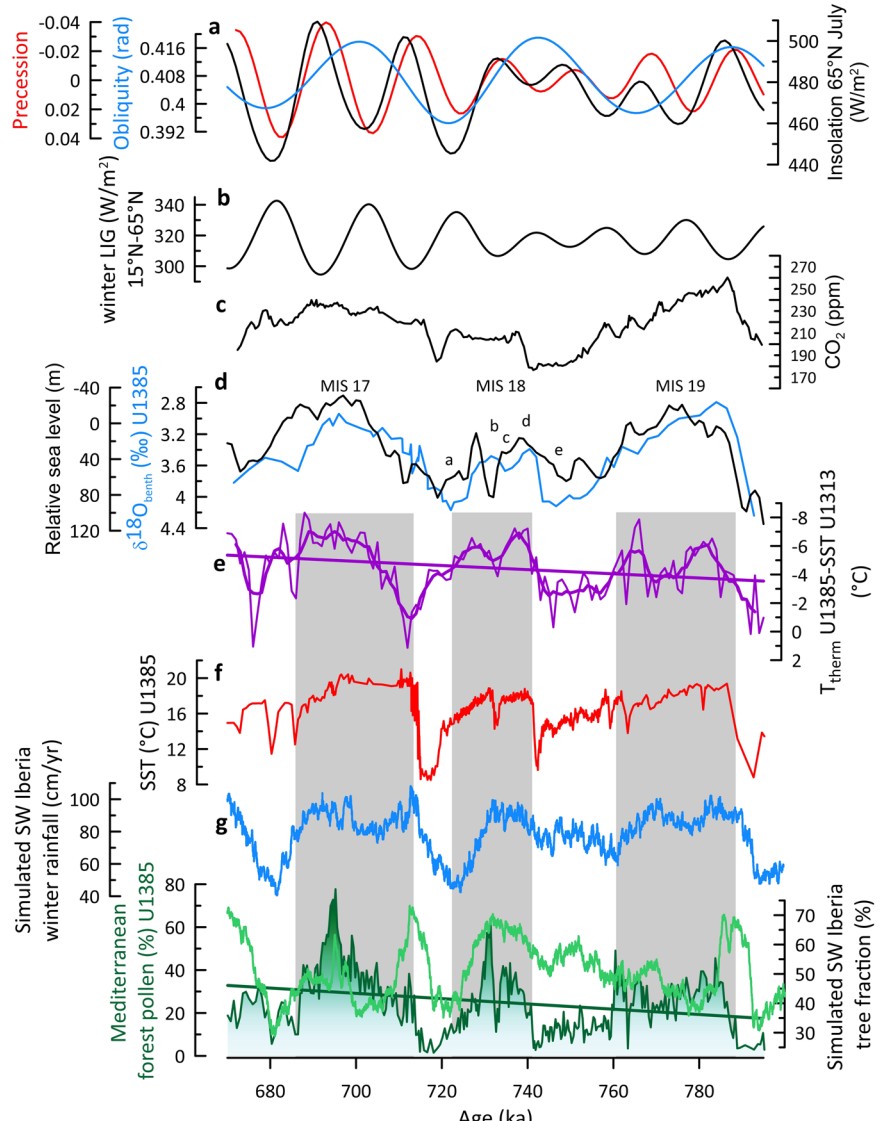

**Fig. 3 | Global and Eastern North Atlantic paleoclimatic records for the interval Marine Isotope Stage (MIS) 19-17. a** insolation at 65°N in July (black), obliquity (blue) and precession (red)[26], **b** winter Latitudinal Insolation Gradient (LIG) between 15°N and 65°N (this study), **c** atmospheric $CO_2$ concentrations[28], **d** $\delta^{18}O$ of benthic foraminifera from IODP Site U1385 (blue)[30], a-e: sub-stages of MIS 18[80], and relative sea level changes from ODP Site 1123 (ref. 29) compared to present-day (black), **e** temperature gradient between T thermocline from IODP Site U1385 (37°N) and sea surface temperature (SST) from IODP Site U1313 (41°N)[3], **f** alkenone-based SST from IODP Site U1385 (this study), **g** simulated South-West Iberia winter rainfall (blue), Mediterranean forest pollen percentages (green) from IODP Site U1385, and simulated South-West Iberia tree fraction percentages (light green) (this study). Panels **e**, **g** straight lines indicate ordinary least squares fits. Panel **e** thick line indicates weighted average fit with a 5-sample window. Grey bands indicate the forested phases that define the terrestrial interglacials.

concentration, is marked by limited forest expansion[32] indicating lower winter precipitation compared to MIS 17. More interestingly, the MIS 18 glacial was more forested, reflecting stronger winter rainfall, compared to the preceding MIS 19 despite that the latter interglacial was characterised by higher insolation, sea level, atmospheric $CO_2$ concentrations and similar warm SST compared to MIS 18 (Fig. 3). Previous paleoceanographic studies have shown that prior to MIS 16 the North Atlantic deep-water formation, releasing heat and humidity to the atmosphere and controlling the global climate, was located further south and west (Boreal heat pump) compared to the period after MIS 16 (Nordic heat pump)[33]. This position allowed the arrival of a substantial amount of moisture to southern Europe during the interval MIS 19-17 but does not explain the wetter conditions during the MIS 18 glacial compared to the MIS 19 interglacial.

To ground-truth the proxy-based evidence, we utilised simulated annual and seasonal air temperatures and precipitation as well as tree fraction from MIS 19 to 17 generated with the iLOVECLIM model solely forced with changes in insolation, ice-sheet reconstruction and greenhouse gas concentrations (Figs. 3, 4 and S6). The numerical model data in fact show high levels of winter rainfall for the three MISs and similar tree fraction percentages during MIS 18 and MIS 19 as inferred from the pollen data. In contrast to the proxy data, the simulated tree fraction is the weakest during MIS 17 (Fig. 3). The simulated winter rainfall is the highest during MIS 17 but shows an overall good agreement with the simulated tree fraction during MIS 18 (Fig. 3). This mismatch between model and proxy reconstructions could be explained by the difficulty in quantitatively estimating the forest cover from pollen data[15]. However, our approach identifies substantial qualitative forest cover and winter rainfall changes that tightly parallel changes in the $T_{therm}$ SST gradient (Fig. 3). Thus, the mismatch could be as well the result of a feedback process that is not well reproduced in iLOVECLIM such as the poor

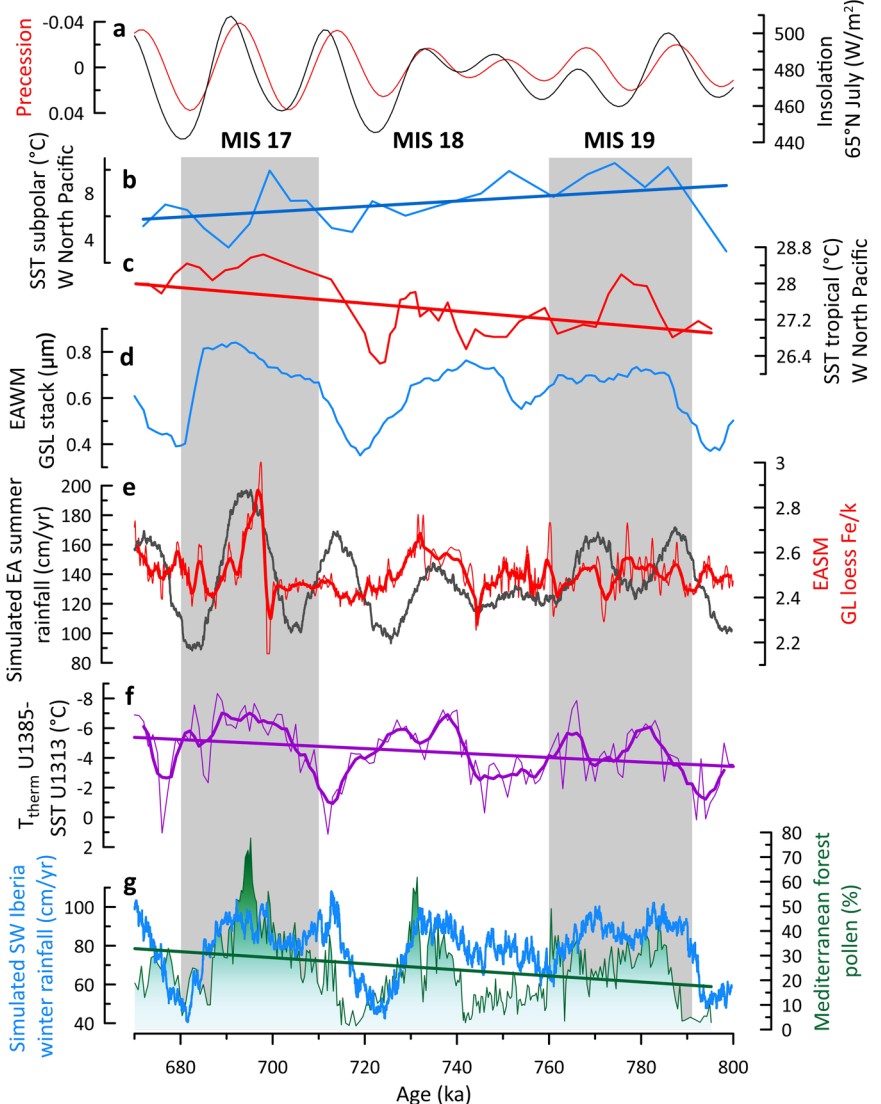

**Fig. 4 | Paleoclimatic records from Eastern North Atlantic margin and the Chinese Loess Plateau compared to subpolar and tropical West North Pacific sea surface temperature (SST) records for the interval Marine Isotope Stage (MIS) 19-17. a** insolation at 65°N in July (black), and precession (red)[26], **b** SST record from ODP Site 882 located in the subpolar West North Pacific Ocean[14], **c** SST record from ODP Site 1143 located in the tropical West North Pacific Ocean[13], **d** Grain Size Loess (GSL) record from the Chinese sequences of Pingliang, Jingshuan, Baoji, Lingtai, Puxian indicating changes in East Asia winter monsoon (EAWM)[11], **e** Fe/K record from the Chinese Loess Plateau Gulang sequence indicating changes in East Asia summer monsoon (EASM) (red)[10], and iLOVECLIM simulated summer rainfall (grey, this study–the data have been smoothed using a 10-year running mean), **f** temperature gradient between T thermocline from IODP Site U1385 (37°N) and SST from IODP Site U1313 (41°N)[3], **g** Mediterranean forest pollen percentages (green) from IODP Site U1385, and simulated South-West Iberia winter rainfall (blue) (this study). Panels **b, c, f, g**: straight lines indicate ordinary least squares fits. Panel **e, f**: thick lines indicate weighted average fit with a 23- and 5-sample windows, respectively. Grey bands indicate the MIS interglacials 19 and 17.

prediction of the ocean thermal gradient despite a robust SST simulation.

In order to identify the main driver that may explain the increasing trend of forest cover and winter rainfall from MIS 19 to MIS 17, and particularly more rainfall during MIS 18 glacial compared to MIS 19 interglacial, we investigate four large-scale reconstructed climate variables characterising the eastern North Atlantic (Fig. 3): (a) the amount of winter precipitation using the Mediterranean forest pollen percentage record (this study), (b) the position and intensity of the westerlies using the winter latitudinal insolation gradient as proposed by Davis and Brewer[34], (c) the position of the North Atlantic moisture source using the $T_{therm}$ SST gradient[3], and (d) the growth and decay of the northern hemisphere ice-sheets using the benthic foraminifera $\delta^{18}O$ record from Site U1385 (ref. 30) that strongly parallels the LR04 $\delta^{18}O$ stack curve[35]. Since large-scale reconstructed climate time series

are unevenly spaced, we used the binned correlation technique[36,37] to convert the unevenly spaced paleoclimate time series to evenly spaced time series (Fig. S4). To estimate the time-dependence of the strength of association between the large-scale reconstructed time series and the Mediterranean forest pollen (MF), we used a robust Pearson correlation approach that takes into account the serially dependence of the time series under study[38] (Fig. S4).The Pearson's correlation analysis shows a high indirect correlation, with a correlation coefficient ("r") statistically significant of −0.538 [−0.735; −0.258], between the Mediterranean forest and the $T_{therm}$-SST gradient record for the whole studied interval, MIS 17, 18, and 19. The correlation is also statistically significant and slightly higher, −0586 [−0.781; −0.288] when MIS 18 is taken individually. This indirect correlation indicates that the northward displacement trend of the ENACW, weakening the $T_{therm}$ SST gradient trend from MIS 19 to MIS 17, correlates well with the

long-term Mediterranean forest increase, i.e., winter rainfall, over this interval. An indirect correlation also exists between Mediterranean forest and ice-volume changes with a $r$ of $-0.623$ [$-0.829$; $-0.268$]. No statistical significant correlation was found ($r = -0.178$ [$-0.525$; $0.220$]) between Mediterranean forest cover and winter latitudinal insolation gradient. The progressive increase in warmer and wetter conditions in South-West Iberia, induced by the northward latitudinal shift of the oceanic moisture source to the Iberian margin, appears to be a distinctive feature of South-West Europe at the end of the MPT. Simulated relative humidity, i.e., the moisture content in the lowest atmospheric layer (at 800 hPa), between Sites U1313 and U1385 shows the northward migration of the moisture source from MIS 19 to MIS 18 although it does not reproduce the northern position of the moisture source during MIS 17 (Fig. S5) suggested from marine and pollen data. Pollen records from Albania/North Macedonia and North-East Greece show that, while high temperate forest cover characterised South-East Europe during MIS 18[39,40], no linear increasing forest cover trend is identified in this region from MIS 19 to MIS 17 (Fig. S6).

## Contemporaneous strong moisture production in the North Atlantic and West North Pacific during the MIS 18 glacial

Interestingly, the long-term trend towards wetter winter conditions recorded in South-West Iberia from MIS 19 to 17 (Fig. 4) parallels the progressive increase in the intensity of the East Asian summer monsoon (EASM) (Fig. 4), as inferred from the Fe/K ratio, a proxy for summer monsoon-induced changes, in the Gulang loess section (37° 30′ N, 102° 53′ E, North-West margin of the Chinese Loess Plateau, Fig. 1)[10], both regions being located at similar subtropical latitudes (~35–37°N). The δ[13]C of the loess carbonate from the Jingyuan sequence (36.35°N, 104.6°E) also shows summer precipitation during MIS 18 as high as during MIS 19[41], also indicating the interglacial character of MIS 18 glacial. The EASM was as strong or stronger during the weaker summer insolation and more glaciated MIS 18 period than during the MIS 19 interglacial, and weaker compared to MIS 17 (Fig. 4). The intensity of the East Asian winter monsoon (EAWM), marked by cold and dry outbursts in the Chinese Loess Plateau related to the development of Siberian Highs[42], was similar or more intense during MIS 18 compared to MIS 19 as shown by the mean grain size of quartz particles (MGSQ)[12] and grain size loess (GSL) stack records from the Chinese Loess Plateau[11] (Fig. 4 and S6). Simulated summer rainfall in East Asia shows an increase in EASM during MIS 18 although weaker compared to the reconstructed loess-based EASM (Fig. 4). The amount of oceanic moisture production in the tropical North Pacific across the MIS 19-17 interval could explain the observed long-term wetting trend in East Asia. The intense EAWM recorded during MIS 18 was associated with SST cooling in the high latitudes[14] and strong warming in the tropical sector[13] (Fig. 4 and S6) causing an increase of the meridional SST gradient in the West North Pacific Ocean. This SST gradient increase may lead to the strong production of water vapour in the Indian and Pacific tropical sectors enhancing the EASM over North-East China[43] as reflected by the Gulang and Jingyuan sections.

At present, a strong SST gradient exists off the coast of East Asia during the early summer monsoon as the result of the previous winter's cold air outbreaks increasing rainfall over southern Taiwan (24°N)[44]. During the midsummer, the peak westerlies migrate north of the Plateau, the extratropical northerlies weaken, leaving only the monsoon low-level circulation that penetrates North-East China, which, coupled with stronger monsoonal southerlies, leads to the northward migration of the rain band[44–46]. The stronger EASM during MIS 18 compared to MIS 19 in North China would be the result of the penetration of higher amounts of oceanic moisture into East Asia channelled by the northward migration of the westerlies. The simulated relative humidity between ODP Sites 882 and 1143 shows indeed a northward position of the moisture source during both MIS 19 and MIS 18 (Fig. S5). During the MIS 18 glacial, the strong summer precipitation

in East Asia was contemporaneous with the high moisture production and winter rainfall recorded in the subtropical latitudes of the North Atlantic, an atmospheric configuration that is compatible with the strong development of the Siberian Highs[42].

During MIS 17, interglacial conditions associated with higher insolation would produce the intensification of the SST gradient in the West North Pacific and Indian tropical sectors and, therefore the increase of the EASM amplified by the contemporaneous increase of cold outbursts (Fig. 4), compared to MIS 18. The stronger winter rainfall in South-West Europe during MIS 17 compared to MIS 18 (Fig. 3 and S6) from both pollen and climate simulation would be the result of the nearest position of the ENACW subduction centre to South-West Iberia caused by the progressive increase of the northward transport of heat and moisture by the North Atlantic Current into the Nordic Seas associated with the establishment of the Nordic heat pump after ~700 ka[4,33].

These results reveal for the first time the paradoxical interglacial character of MIS 18 glacial in the Eurasian subtropical latitudes, and the role of the oceanic moisture production in explaining the strong precipitation during a glacial period. Not only the direction and the intensity of the westerlies but the amount of oceanic moisture production should be taken into account to explain the recorded evolution of South-West European precipitation and EASM. Furthermore, sensitive model experiments show that during the intervals marked by northern ice-sheets of intermediate size, such as our studied interval, the ice growth rate will potentially increase with the strengthening of the North Atlantic Current as summer temperature remains at 0 °C at the south of the major ice-sheets while precipitation increase over this area at least in the Eurasian sector[4]. Therefore, the nearly continuous moisture production in the North Atlantic and West North Pacific subtropical sectors and its progressive northward transport by the westerlies during the end of the MPT may have substantially contributed to the strongest glaciation of the last millions of years in Eurasia and North America during MIS 16, leading to the strong dominant 100,000-year ice age cycles. This process could also be at work in previous moderate glacial/interglacial periods of the MPT. The long-term increasing $T_{therm}$ SST gradient trend in the North Atlantic from MIS 25 to 22 is similar to that observed from MIS 19 to 17[3] and could have led to the remarkable expansion of the Northern Hemisphere ice during MIS 22.

## Methods

### Environmental setting of IODP Site U1385

Site U1385 was recovered during IODP Expedition 339 "Mediterranean Outflow". The site is located on a spur, the Promontorio dos Principes de Avis, along the continental slope of the South-West Iberian margin, which is elevated above the abyssal plain avoiding the influence of turbidites[30]. The water depth of Site U1385 places it under the influence of Northeast Atlantic Deep Water today, although it was influenced by southern sourced waters during glacial periods[47].

The South-West Iberian margin is located in the north-eastern edge of the subtropical gyre, under the influence of Eastern North Atlantic central water (ENACW). The surface water column is affected by the Portugal current (PC), which brings cold nutrient-rich water from the northern latitudes and forms the ENACW of subpolar origin (ENACWsp), and by the Azores current (AC) which brings warm water from the Azores front generating the ENACW of subtropical origin (ENACWst)[48]. The general distribution of water masses is influenced by the seasonal migration of the Azores anticyclonic cell and its associated large-scale wind pattern.

Climate in South-West Iberia is directly affected by the intensity and direction of the westerlies that are, in turn, controlled by the North Atlantic Oscillation (NAO)[49,50]. During winter the North Atlantic westerlies bring moisture to South-West Iberia, while a high-pressure cell develops in the North Atlantic during summer, which generates strong northerly trade winds inducing coastal upwelling[51]. This climate

seasonality is characterised by wet and mild winters (Tmin: 5–1 °C; Tmax: 13–8 °C) and hot and dry summers, annual precipitation <600 mm[52], and leads to the development of a Mediterranean vegetation in the adjacent landmasses dominated by deciduous oak at middle elevation, and evergreen oak, olive tree, *Pistacia, Phillyrea* and rockroses (*Cistus*) at lower elevations[18]. Ericaceae (heathlands) are restricted to rainy mountains (annual precipitation >600 mm), strong oceanic conditions and locations with wet soils[53].

## Stratigraphical framework

The sedimentary section recovered at Site U1385 (150 m-long composite record) shows hemipelagic continental margin sediments deposited under normal marine conditions with a fully oxygenated water column and average sedimentation rates of 10 cm/ky[54]. The stratigraphy of Site U1385 was built upon a combination of chemo-stratigraphic proxies[30]. Ca/Ti ratio measured every cm in all holes by core scanning X-ray fluorescence (XRF) was used to construct a composite section, and the low resolution (20 cm) benthic foraminifera oxygen isotopic record ($\delta^{18}O_{benth}$) was correlated to the marine $\delta^{18}O_{benth}$ stack of LR04[35] to provide the age model that we present here[30].

## Pollen-based vegetation and climate reconstruction

Sediment subsamples of 1-cm thickness and 1.5–5 cm³ volume were prepared for pollen analysis using the standard protocol for marine samples (https://www.epoc.u-bordeaux.fr/index.php?lang=fr&page=eq_paleo_pollens), employing coarse-sieving at 150 µm, successive treatments with cold HCl, cold HF at increasing strength and microsieving (10 µm mesh). Known quantities of *Lycopodium* spores in tablet form were added to permit the calculation of pollen concentrations. Slides were prepared using a mobile mounting medium to allow rotation of the grains and counted using a Primo Star light microscope at 400 and 1000-fold magnifications for routine identification of pollen and spores, respectively. One hundred twenty-four samples were analysed every 4 cm (450-yr average temporal resolution). Pollen counts oscillate between 100 and 152 terrestrial pollen grains excluding *Pinus* (main pollen sum), aquatics and spores. The main pollen sum plus *Pinus* oscillates between 205 and 1438. The number of pollen morphotypes in most of the samples, 98 samples out from 124, ranges from 20 to 27, and from 16 to 19 morphotypes in the remaining samples. Pollen percentages for terrestrial taxa were calculated against the main sum of terrestrial grains, while percentages for *Pinus* were calculated against the main sum plus *Pinus*. Aquatic pollen and spore percentages are based on the total sum (Pollen + spores + indeterminables + unknowns) (Fig. S1). The uncertainties of the pollen percentages of the main ecological groups at 95% have been calculated using the "exactci" function in the PropCIs v. 0.3.0 package (Package "PropCIs"). The calculation of CIs is based on the 'exact' Clopper–Pearson method assuming a binomial proportion[55,56]. The average uncertainty of the calculated pollen percentage of the Mediterranean forest values in our analysis is less than 8% (Fig. S2). Total spores and pollen concentrations oscillate between 2000 and 28,000 grains.cm⁻³ (Fig. 2). Changes in grain concentrations do not parallel changes in pollen percentages and, therefore, these latter changes indicate actual variations in forest cover and composition. The interpretation of the pollen diagram was assisted by a constrained hierarchical cluster analysis based on Euclidean distance between samples (Fig. S1). Analysis was performed in the R environment v. 3.6.3 using the "chclust" function from the *Rioja* package[57]. This analysis identifies three main pollen zones that we interpret as a strong expansion of the forest cover bracketed by two open vegetation phases.

## Sea surface temperature reconstruction and identification of freshwater pulses

Sea Surface Temperature (SST) was reconstructed using di- and tri-unsaturated alkenones of 37 carbons atoms, which are organic compounds synthesised by marine coccolithophore algae in a temperature-related proportion. Alkenones, in particular the tetra-unsaturated compounds ($C_{37:4}$), can also be used to track episodes of massive cold freshwater input (from iceberg melting or river discharges), which are responsible for decreasing salinities in the surface water masses[8,58]. Alkenones are part of the total lipid extracted (TLE) fraction which can be extracted from 2 g of sediment by sonication with dichloromethane and hydrolysed with 6% potassium hydroxide in methanol. After derivatization with bis(trimethylsilyl)trifluoroacetamide, the TLE was identified using a Bruker mass spectrometer detector and quantified with a Varian gas chromatograph Model 3800 equipped with septum programmable injector and a flame ionisation detector with a CPSIL-5 CB column. As a gas carrier was used hydrogen at 2.5 ml.min⁻¹. Alkenone concentrations were determined using n-hexatriacontane as an internal standard. Reproducibility tests showed that uncertainty in the alkenone index $U^k_{37}$ determinations were lower than 0.015 (ref. 59). The $U^k_{37}$ was calculated based on the di- and tri- unsaturated concentrations[60], and converted into temperature values using the global core top calibration of annual SST[61]. The average temporal resolution between samples is 528 years and the uncertainty of SST reconstruction is ±0.5 °C (ref. 62).

## Data analyses

To correlate the paleoclimate records, we used first the binned correlation technique[36,37] to convert the unevenly spaced paleoclimate time series under analysis to evenly spaced time series. This technique is based on a novel estimation approach proposed by ref. 36 for estimating the correlation between two paleoclimate time series with different timescales. The idea is that autocorrelation means that memory enables values obtained on different time points to be correlated. Binned correlation is performed by resampling the unevenly spaced paleoclimate time series under analysis into time bins on a regular grid, assigning the mean values of the variable under scrutiny within those bins. This method was recently implemented in the R package BINCOR[63], which is freely available on CRAN. To estimate the time-dependence of the strength of association between the large-scale reconstructed time series and the Mediterranean forest pollen (MF), we used a robust correlation approach that takes into account the serially dependence of the time series under study. We have followed the method developed by[38] to estimate the Pearson's correlation coefficients that takes into account the serially dependence. This method also includes an estimation of a confidence interval (95%) obtained through a nonparametric stationary bootstrap technique with an average block length proportional to the maximum estimated persistence time of the data under analysis. This statistical technique is implemented in the software PearsonT3, which is freely available from http://www.climate-risk-analysis.com.

## iLOVECLIM model

We used the intermediate complexity climate model iLOVECLIM in version 1.1.5 (ref. 64), which is a development branch of the LOVECLIM model in version 1.2 described in Goosse et al.[65]. iLOVECLIM does not present major differences in the physics of the atmosphere and ocean compared to LOVECLIM. The based components of this version are the atmosphere ECBilt[66], the ocean CLIO[67] and the vegetation and land surface VECODE[68]. The modelled tree fraction in VECODE includes shrub plants such as Ericaceae[68]. Oxygen isotopes (¹⁸O and ¹⁶O) have been implemented in the coupled model and evaluated against observed data in water samples and carbonates[69]. An equilibrium run was first performed for 5000 years under climatic conditions at 800 ka. We then ran a transient run between 800 and 670 ka. The insolation[70], greenhouse gas concentrations[28,71] and prescribed ice-sheets reconstruction[72] have been updated with an acceleration factor of 10 for the transient simulation. The model has been run at T21 spatial resolution (5.6° in longitude and latitude) and the output

are computed with an annual timestep and with a monthly timestep for the precipitation only. iLOVECLIM was previously successfully applied in the Asian monsoon region to investigate the oxygen isotopic composition of precipitation and calcite in association with monsoonal precipitation changes over the last 150 ka[73]. The $\delta^{18}O$ in precipitation computed in the model is extracted over the East Asia region (~25°N-41.5°N and ~90°E-118°E). The mean $\delta^{18}O_{calcite}$ is then computed from $\delta^{18}O_{precipitation}$, corrected from ice-sheet contribution to the global seawater $\delta^{18}O$ (ref. 74) and from atmospheric temperature at 2 m height using the following Eq. (1) for synthetic calcite under equilibrium conditions[75].

$$1000 \ln\alpha\left(\text{calcite} - \text{H}_2\text{O}\right) = 18.03 \left(\frac{10^3}{T}\right) - 32.42, \qquad (1)$$

where $\alpha$ is the fractionation factor and $T$ the temperature in Kelvin. The values are then converted from SMOW scale to VPDB scale using the equation of Coplen et al. [76] (2), and smoothed using a 10-year running mean:

$$\delta^{18}O_{\text{calciteVPDB}} = 0.97002\, \delta^{18}O_{\text{SMOW}} - 29.98. \qquad (2)$$

with 0.97002 the isotopic fractionation factor between VPDB and SMOW, and 29.98 expressed in per mil. The air temperature at 2 m height, annual precipitation and tree fraction are also computed over South-West Iberia region (~36°N-41.5°N and ~5.6°W-11.2°W), which includes most of the entire Tagus basin from where the majority of pollen grains arrive the to the South-West Iberia margin sediments[18].

## Data availability
The elemental data that support the findings of this research are provided in PANGAEA database (https://doi.org/10.1594/PANGAEA. 957610).

## Code availability
Codes for the statistical analyses (Bincor and Pearson's correlations) are available in the Supplementary Information. The iLOVECLIM source code and developments are hosted at http://forge.ipsl.jussieu. fr/ludus (ref. 77) but are not publicly available due to copyright restrictions. Access can be granted on demand by request to D. M. Roche (didier.roche@lsce.ipsl.fr) to those who conduct research in collaboration with the iLOVECLIM user group.

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

## Acknowledgements

This research used samples collected during the Expedition no. 339 "Mediterranean Outflow" of the Integrated Ocean Drilling Program (IODP). M.F.S.G. acknowledges funding from the GPR Human Past (University of Bordeaux). A.B. thanks Deutsche Forschungsgemeinschaft (DFG), project BA 3809/8. C.Z. acknowledges funding from IODP France and J.M.P.-M. from the Junta de Castilla y León and the European Regional Development Fund (Grant CLU-2019-03). T.R. acknowledges funding from FCT through projects Hydroshift (PTDC/CTA-CLI/4297/2021), WarmWorld (PTDC/CTA-GEO/29897/2017), UIDB/04326/2020, UIDP/04326/2020, LA/P/0101/2020 and EMSO-PT (POCI-01-0145-FEDER-022157). We thank Vincent Hanquiez for drawing Fig. 1 and Ludovic Devaux for pollen sample preparation.

## Author contributions

M.F.S.G. designed the research, performed the pollen analysis, and wrote the manuscript, T.E. and J.M.P.M. performed the model simulations and the statistical analyses, respectively. T.R. performed the geochemical analyses (alkenone and %$C_{37:4}$). T.E., J.M.P.M., C.Z., T.R. and A.B. participated in the discussion and final writing of the manuscript.

## Competing interests

The authors declare no competing interests
