## [Peer Review File · Nature Communications]

Moist and warm conditions in Eurasia during the last glacial of the Middle Pleistocene TransitionReviewer #1 (Remarks to the Author):

Sanchez Goni et al., Interglacial climates during the last glacial of the Middle Pleistocene, Nature Communications

Summary: The manuscript presents new data on regional climate (Iberian-Mediterranean region) for the Middle Pleistocene time interval (800 to 670 ka (thousand years before present)). The data are proxy measurements done on a marine sediment core and, according to the manuscript, indicate: sea-surface temperature, forest cover, westerlies (intensity) and winter precipitation. If the data would be correct, then the manuscript would be a major step forward because of the data itself and also their interpretation („interglacial conditions recorded ... during the MIS 18 glacial“, line 26-27) and the manuscript thus would make a good contribution to Nature Communications. However, (A) the proxy quality and proxy error are poorly substantiated by the manuscript and (B) there is not sufficient work contained in the paper on a comparison of the new proxy data with previous findings (same climate variables in similar space-time domains and resolutions). As a methodological weakness, (C) the employed method of interpolation is not necessary for the employment of robust statistical methods (i.e., those that do not require even time spacing) and, further, (D) the interpolation to an equidistant spacing of 250 years is too small bearing in mind that the original pollen series have an average time resolution of 450 years (i.e., if you generate more data via interpolation than there were originally measured, you may get too small uncertainty intervals for your statistical estimations on the data). Finally, (E), the degree of sloppiness (typos, grammatical errors, inconsistent notation, inaccurate formulations, etc.) is beyond what readers or reviewers should expect from Nature Communications.

Recommendation: Major revisions. That is, overcome above mentioned deficits (A to E) by doing the following points.

[1] Test hard the existing proxy knowledge and show the validity of your proxy reconstructions (i.e., via comparison with existing data in overlapping space-time intervals). A comparison with climate-model output is nice but not sufficient (i.e., you have to compare also against real data).

[2] Use instead of/in addition to the wavelet approach the method of binned correlation coefficient that can deliver correlation information of two variables that may be recorded on different and unevenly spaced timescales. The reference is Polanco-Martinez et al. 2019 (reference 59 in manuscript). You may compare binned correlation results with those using wavelet approach, however, the latter obtained using a resolution of 450 years, not 250 years.

[3] Reduce the degree of sloppiness (see the detailed list below).

Detailed points

Line 1

The title phrase („interglacial climates during the [last] glacial“) sounds ugly; maybe use instead of „interglacial climates“ the expression „warm climate stages“ or similar.

Lines 15ff

The abstract should be more clearly written in „Nature style“. In particular, the key references for what is known have to be added.

Line 16

Define the unit „ka“ at its first usage („thousand years before present“).

Line 22

Write „a tracer of northern westerlies“.

Lines 26-27

Again (cf. Line 1), „with interglacial conditions ... during the MIS 18 glacial“ sounds ugly.

Line 31

Remove „(MIS 18)“ from the keyword list.

Line 93

Correct referencing is „As discussed³, in the entire period ...“.

Lines 126-128

It seems too easy to blame just the climate model for the „mismatch between model and proxy reconstructions“. One has to seriously and self-critically discuss also the limitations of the proxy data.

Lines 140-141

„Statistically significant correlation coefficient“: this issue with significance is rather tricky with correlations on data that (A) could show temporal serial dependence or (B) non-Gaussian distributions. Dr. Mudelse, amongst others, has done work on that, which needs to be consulted.

Lines 246-247

The expression „delta18Ob“ is highly unusual in paleoclimate sciences. Perhaps use „delta18O_{benth}“ (i.e., with subscript) to indicate the benthic character.

Line 250

The unit „cm³“ has to be written with a superscripted „3“.

Line 255

The way to express magnifications is grammatically not correct. One can use, for example, „400- and 1000-fold magnification“.

Line 257

The manuscript states that 4 cm depth resolution corresponds to a time resolution of 450 yr, although the given sedimentation rate (line 243) is 10 cm per year. This is numerically at odds.

Line 260

The manuscript mentions „98 samples out from 126“, while before (line 256) it reports about „One hundred twenty-four“. This is at odds.

Line 265

While it is OK to mention the software (here: „exactci“) used for the uncertainty calculation, the manuscript must inform readers about the methodological approach plus which statistical assumptions were made for uncertainty determination (e.g., error propagation assuming Poisson distributions of pollen counts, or bootstrap resampling, etc.).

Line 267

The expression „sporo-pollen“ is not correct.

Line 273

What does the manuscript mean with the wording „robustly identifies“? Robust against which possible violations of made assumptions?

Line 277

Write „atoms“ instead of „atmos“.

Line 279

The right parenthesis in the expression „(C₃₇:4)“ has not to be in subscript.

Line 282

Write „2 g“ (i.e., with a space between number and unit).

Line 284

„Mass spectrometer“ has not to be capitalized.

Line 285

„Gas chromatograph“ has not to be capitalized.

Line 287

„Hydrogen“ is typically not capitalized in scientific texts.

Line 289

You cannot put the reference (54) directly after the numerical value (0.015) in order to prevent misinterpretation (0.015 to the power of 54); instead use „0.015 (ref. 54)“.

Line 289

Insert a space after „Uk'_37“.

Line 292

Write „(ref. 57)“.

Line 321

The expression „Oxygen-18-16 isotopes“ is highly unusual.

Line 331

Write „(ref. 70)“.

Line 332

Write „temperature at 2 m height“.

Line 332-333, the equation

Insert a space after „1000“. Do not write „ln“ in italics. Do not write „H2O“ in italics. Put a comma at the end of the equation.

Lines 335-336, the equation

What does the left subscript „2“ mean? Do not write „calcite VPDB“ in italics. Do not write „SMOW“ in italics. Put a unit to the number (29.98), likely it is „permil“. Put a period at the end of the equation.

Lines 316-338

There is no indication about the temporal resolution of the iLOCECLIM model output.

Line 345

Abbreviate as „J.M.P.-M.“.

Line 432

Write „Scientific Reports“.

Line 473

Reference Stow et al. is not complete.

Line 500

Do not capitalize the title words of the referenced article.

Line 503

Some wrong text („“).

Line 505

Reference Goosse incomplete (author list).

Line 516

Do not abbreviate journal title.

Line 519

The correct journal name is „Journal of the Atmospheric Sciences“.

Line 525

Do not abbreviate journal title.

Lines 545-550

„LO“ and „TP“ not to be defined since spelled out on figure graphics. Give more details about the „westerlies“: applies for recent and/or paleo?, perhaps stretches much farther north than shown in figure?, and msiing is a reference for paleo westerly position.

Figure 1

Give color scale bar (bathymetry, height?). Why are the EAWM arrows left-circling and not right-circling? Place is on northern hemisphere. The abbreviations (e.g., ENACW) are way too difficult to discern: use a uni-color background for those text snippets in the graphics figure.

Line 570-572

„MIS“ is already defined. The colon in „MIS :Marine“ is misplaced. Do not capitalize „Relative“, „Atmospheric“, „Winter“ or „Insolation“.

Figure 2

Give units of obliquity. Employ correct ordering of panel labels: from top (a) to bottom (g), not vice versa. „Relative Sea level“: do not capitalize „Sea“. Precession-y-axis to be shifted to the right for better legibility. „[CO₂] ppm“ to be changed into „CO₂ (ppm)“. Subscript index „therm“ too small font size. Panels a, c: define straight lines (e.g., „ordinary least squares straight line fits“). Panel c: define thick and thin lines (likely something with smoothing).

Figure 3

Use correct panel labels (i.e., not in boxes). Write unit „ka“ instead of „Ka“. Write „Age (ka)“ instead of „Years BP (Ka)“.

Figure 4

Write unit „ka“ instead of „Ka“.

Reviewer #2 (Remarks to the Author):

This paper presents paired changes in SST and pollen-based forest cover from IODP site U1385 spanning the MIS 19-17, aiming to address the potential role of enhanced moisture supply from mid-latitude oceans to higher latitudes in triggering the final onset of significant 100-ka ice cycles. More specifically, the authors propose that MIS 18 is characterized by relatively warm and wet glacial that favors for sustained ice accumulation in the southern Eurasian. Statistical and model results suggest that SST gradient is the likely driver for increasing trends of winter precipitation and forest cover from MIS 19 to 17 on both orbital and millennial timescales. By comparing with East Asian summer monsoon proxies from Chinese loess, the authors conclude that the moisture production in the mid-latitude oceans over the Northern Hemisphere was increased and progressively transported northward during moderate MIS18 glacial, which may result in the full development of the dominant 100-ka ice cycles. The hypothesis is very interesting for better deciphering why the MPT ended around 700 ka. But I have several specific concerns that needs further clarification.

1. During MPT, there are many mild interglacial periods (i.e. MIS23, 27, 33) similar to the climate background of MIS 18 (see Hodell et al., 2022, Clim Past.). The MIS 23 is also followed by a remarkable expansion of the Northern Hemisphere ice during MIS 22. I just wonder whether the authors' hypothesis is also workable in moderate glacial/interglacial periods during the MPT.

2. Millennial-to-centennial variability. Rapid changes in forest pollen concentration are amazing (Fig.2a and 4a). Unfortunately, this point is only mentioned in lines 150-155). Not sure whether the mechanisms of coupled winter precipitation and SST gradient on both orbital and millennial timescales are similar not.

3. Simulated rainfall and tree fraction based on iLoveclim model are compared with the proxies from

the U1385 (Fig.2a and 4a). It seems that the amplitudes of model- and pollen-based forest changes are visually different. This amplitude mismatch needs further clarification. If the authors propose that the forest pollen is related to the winter precipitation, I would suggest to add the simulated winter rainfall in Fig.4a for direct comparison.

4.Loess proxies are used to verify intensified summer monsoon and moisture transport during the MIS 18 (Fig.4 and lines 164-182). These loess proxies are from northwest China (not NE China) and the $\delta^{13}\text{C}$ of loess carbonate (Sun et al., 2019, Nat Commun) might be more sensitive to summer monsoon change than the Fe/K ratio. My concern is that in East Asian, enhanced precipitation is occurred in the summer season. How to differentiate the roles of enhanced winter/summer moisture production in mid-latitude North Atlantic/northwest Pacific in accumulating the ice expansion at the end of the MPT.

The authors suggest that the oceanic moisture production plays a key role in explaining the strong winter precipitation during a glacial period (lines 202-206). I wonder whether iLOVECLIM simulation results can demonstrate this enhanced moisture transport during MIS 18 relative to MIS 19. The differences in moisture transport flux and precipitation between MIS 18 and 19 can be inferred from the transient modeling results.

Reviewer #3 (Remarks to the Author):

Dear authors,

I think that your paper titled "Interglacial climates during the last glacial of the Middle Pleistocene Transition" is a good one and that the data you use (even if already partly published) allow new advances in the knowledge of the MPT climate.

I have one main concern:

- How can you state that the expansions of deciduous oaks are all components of the Mediterranean forest? This idea risks to be an exaggeration of the principle of actualism. We well know that during the Mid Pleistocene Transition the vegetation was quite different from the present one, with the presence of many arboreal species growing in wet environments, that are at present extinct in Europe. Deciduous oaks pollen could therefore include also oaks living in wet environments and not only in relative dry conditions. This wider interpretation would fit more with the Balkan terrestrial records (Tenaghi Philippon and Ohrid) in which deciduous oaks are included in the mesophilous vegetation.

I include the original files with many comments and minor issues, please consider them.

There are also minor issues in the figures (e.g. red and green used in the same curve) that should be adjusted.

In their review of the first version of this manuscript, reviewer #3 added some comments to the manuscript files. These comments were forwarded to the authors, who replied as included in this Peer Review File.

Responses to the referees

We address below the suggestions of change made by the three referees on a one to one basis (in blue).

Reviewer #1 (Remarks to the Author):

Sanchez Goni et al., Interglacial climates during the last glacial of the Middle Pleistocene, Nature Communications

Summary: The manuscript presents new data on regional climate (Iberian-Mediterranean region) for the Middle Pleistocene time interval (800 to 670 ka (thousand years before present)). The data are proxy measurements done on a marine sediment core and, according to the manuscript, indicate: sea-surface temperature, forest cover, westerlies (intensity) and winter precipitation. If the data would be correct, then the manuscript would be a major step forward because of the data itself and also their interpretation („interglacial conditions recorded ... during the MIS 18 glacial“, line 26-27) and the manuscript thus would make a good contribution to Nature Communications. However, (A) the proxy quality and proxy error are poorly substantiated by the manuscript and (B) there is not sufficient work contained in the paper on a comparison of the new proxy data with previous findings (same climate variables in similar space-time domains and resolutions). As a methodological weakness, (C) the employed method of interpolation is not necessary for the employment of robust statistical methods (i.e., those that do not require even time spacing) and, further, (D) the interpolation to an equidistant spacing of 250 years is too small bearing in mind that the original pollen series have an average time resolution of 450 years (i.e., if you generate more data via interpolation that there were originally measured, you may get too small uncertainty intervals for your statistical estimations on the data). Finally, (E), the degree of sloppiness (typos, grammatical errors, inconsistent notation, inaccurate formulations, etc.) is beyond what readers or reviewers should expect from Nature Communications.

Recommendation: Major revisions. That is, overcome above mentioned deficits (A to E) by doing the following points.

[1] Test hard the existing proxy knowledge and show the validity of your proxy reconstructions (i.e., via comparison with existing data in overlapping space-time intervals). A comparison with climate-model output is nice but not sufficient (i.e., you have to compare also against real data).

To the best of our knowledge, all the relevant data for our study on the interval MIS 19-17 from Eurasia and the North Atlantic and North Pacific Oceans are included in the manuscript and compared with our new data. We implicitly refer to all the North Atlantic SST records presented and discussed in Barker et al. (2021) showing warm sea surface temperatures (SST) in the North Atlantic mid-latitudes during this interval, and particularly during MIS 18. We have added these sites to the map in Figure 1. We have only found an additional loess sequence from the Chinese Loess Plateau close to the other sequences discussed in the manuscript, the Jingyuan sequence. The $\delta^{13}\text{C}$ of the loess carbonate from this sequence also show summer precipitations during MIS 18 as high as during MIS 19 (Sun et al., 2019, Nat. Comm.), indicating the interglacial character of MIS 18 glacial. We have included this new sequence in the discussion.

[2] Use instead of/in addition to the wavelet approach the method of binned correlation coefficient

that can deliver correlation information of two variables that may be recorded on different and unevenly spaced timescales. The reference is Polanco-Martinez et al. 2019 (reference 59 in manuscript). You may compare binned correlation results with those using wavelet approach, however, the latter obtained using a resolution of 450 years, not 250 years.

Binned correlation is a good tool to estimate correlation between two irregular time series (we have a paper about this topic and it is cited in the MS and by the Reviewer 1, i.e., Polanco-Martínez et al. (2019)). Binned correlation was originally developed by M. Mudelsee (2010, 2014), see Ref. Mudelsee's book). However, we think it is not the best tool to achieve one of the aims of our paper: (1) binned correlation is only available for bivariate analysis and we need a multivariate technique to estimate the relationships between the paleoclimate time series analyzed; and (2) we need a multivariate statistical technique to provide which variable/s play/s the main role of the three variables analyzed to explain the Mediterranean forest (MF) percentages, and the wavelet local multiple correlation (WLMC) is an adequate tool to carry out this aim although it is important to point out that time series analyzed via WLMC must be equally spaced (regular) in time. The WLMC has been recently introduced to analyze multivariate (paleo)climate time series (Polanco-Martínez et al. 2020), it is also an adequate mathematical tool to analyze multiple components (time series) of nonlinear dynamical systems, such as the climate (Polanco-Martínez 2023, forthcoming), and we also recently produced an R package named VisualDom available on CRAN (<https://cran.r-project.org/package=VisualDom>; and its respective paper is in under review; Polanco-Martínez (2023), SoftwareX) to include this quantitative and visual tool. Moreover, we have applied binned correlation through our R package BINCOR (Polanco-Martínez et al. 2018; <https://cran.r-project.org/package=BINCOR>) to our paleoclimate irregular time series, but the number of elements of the new regular (binned) time series is very small and correlations estimated are not reliable.

On the other hand, we agree with the Reviewer 1 that 450 years (i.e. close to the mean value of the sampling time) is a good choice as a sampling time resolution. In Figure 1 we show the histogram of the differences of the sampling times (in ka) for the Mediterranean forest percentages. It is observed that a great number of differences (approx. 73%) are between 200 and 500 years, and the mean value of all differences is 410 years.

Figure 1 - Differences between sampling times (ka) for the Mediterranean forest (MF) pollen %. The number of elements of the MF is 310.

For this reason, we have interpolated the MF each 410 years and we have visually compared the irregular time series with its corresponding regularly interpolated time series (Figure 2). In our view, both interpolated MF time series are acceptable (see Figure 2) although **for this revised version of our MS** we use the paleoclimate time series interpolated each 410 years as it is proposed by Reviewer #1.

Figure 2 - Comparison between the irregular time series with its corresponding regularly interpolated time series for 0.25 ka (left) and 0.41 ka (right).

We have estimated the WLMC for the MF time series and the other three variables interpolated each 410 years and 250 years (Figure 3a and 3b, respectively), the main dominant variable for the centennial and millennial scales (first 2 or 3 time-scales; please note that time-scales for both heat maps are different and these cannot be compared time-scale by time-scale directly) is the “T therm SST gradient” (“GRAD”) for both WLMC figures (left and right). It is particularly interesting that for both WLMC heat maps and for the first and second time-scales GRAD is the dominant variable for the interval 740-720 ka (within MIS 18). Therefore, we could conclude that our WLMC (interpolated each by 410 or 250 years) are quite similar and support our results from a statistical point of view. We have modified Figure 3 in the original manuscript (Figure 4 in the revised manuscript) accordingly. The legend of this figure now explains our choice of the 410 years interpolation. This choice is illustrated by Figure 1 that now is included as Figure S4 in the Additional information. We have added the following paragraph in the Methods section:

“The choice of 410 years as sampling time resolution for the time series analysis is equal to the mean value of the sampling time. In Figure S4 we show the histogram of the differences of the sampling times (in ka) for the Mediterranean forest percentages. It is observed that a great of number of differences (approx. 73%) are between 200 and 500 years, and the mean value of all differences is 410 years.”

Figure 3a – Wavelet local multiple correlation (WLMC) for the variables analyzed interpolating each 410 years.

Figure 3b - Wavelet local multiple correlation (WLMC) for the variables analyzed interpolating each 250 years.

Polanco-Martinez, JM*, Medina-Elizalde, MA, **Sánchez Goni, MF**, Mudelsee, M. (2019). BINCOR: An R package for Estimating the Correlation between Two Unevenly Spaced Time Series. *R Journal*, 11(1), 170-184, <https://journal.r-project.org/archive/2019/RJ-2019-035/index.html>

Polanco-Martínez, JM*, Fernández-Macho, J., & Medina-Elizalde, M. (2020). Dynamic wavelet correlation analysis for multivariate climate time series. *Scientific Reports*, 10(1), 21277. <https://www.nature.com/articles/s41598-020-77767-8>

Polanco-Martínez, JM*. (2023, forthcoming) A computational and graphical approach to analyze the dynamic wavelet correlation among multiple components of a nonlinear dynamical system. *Journal of Applied Non-linear Dynamic*. A preprint version of this paper is available from: https://www.researchgate.net/publication/364358510_A_computational_and_graphical_approach_to_analyze_the_dynamic_wavelet_correlation_among_components_of_a_nonlinear_dynamical_system

Mudelsee, M. (2010, 2014). *Climate time series analysis*, Ed Springer, <https://link.springer.com/book/10.1007/978-3-319-04450-7>

[3] Reduce the degree of sloppiness (see the detailed list below).
Detailed points

Line 1

The title phrase („interglacial climates during the [last] glacial“) sounds ugly; maybe use instead of „interglacial climates“ the expression „warm climate stages“ or similar.

We have modified the title as follows: « Interglacial climates in Eurasia during the ultimate glacial of the Middle Pleistocene transition ».

We think that the use of « warm climate stages » can bring to confusion and weaken the actual message of the manuscript. Since the definition of Interstadial by Jessen and Milthers in 1928, it has become customary to use interglacial and interstadial as terms to define particular types of non-glacial climatic conditions in temperate Europe, as indicated by vegetational changes. Therefore, warm climate stage can refer to Interglacial, i.e. defined from the complete vegetation succession of the temperate and Mediterranean forests (Van der Hammen et al., 1971), or Interstadial, defined as a period which was either too short or too cold to permit the development of temperate deciduous forest of the interglacial type in the region concerned (West, 1972). Our data show all the phases characterizing the succession of vegetation observed during Interglacial periods in the Mediterranean region (Cupressaceae and *Betula* pioneer woodlands / deciduous and evergreen *Quercus- Olea- Phillyrea-Pistacia* forest/deciduous and evergreen *Quercus* woodlands) (Figure S1). In the revised version of the manuscript (lines 91-95), we have added this information to support the fact that our data clearly define an interglacial climate in the Mediterranean region.

For the East Asian Summer monsoon, loess data indicate similar or higher values of precipitation during the MIS 18 glacial compared to the MIS 19 interglacial. For this reason, we define an interglacial climate in N China during MIS 18.

Lines 15

The abstract should be more clearly written in „Nature style“. In particular, the key references for what is known have to be added.

Nature Communications guidelines clearly state that the Abstract must be unreferenced.

Line 16

Define the unit „ka“ at its first usage („thousand years before present“).

Done

Line 22

Write „a tracer of northern westerlies“.

Done

Lines 26-27

Again (cf. Line 1), „with interglacial conditions ... during the MIS 18 glacial“ sounds ugly. Based on the actual definition of Interglacial explained above, we have replaced the original sentence with « interglacial climates in both subtropical regions recorded during the MIS 18 glacial”.

Line 31

Remove „(MIS 18)“ from the keyword list.

Done

Line 93

Correct referencing is „As discussed³, in the entire period ...“.

Done

Lines 126-128

It seems too easy to blame just the climate model for the „mismatch between model and proxy reconstructions“. One has to seriously and self-critically discuss also the limitations of the proxy data. This is a good point and we acknowledge referee 1. A quantitative estimation of forest cover changes from pollen data would allow us a more reliable comparison with the simulated tree fraction. A relatively new approach has been recently developed for Europe by Zanon et al., 2018, based on the coupling of modern terrestrial pollen samples with the corresponding satellite-based forest cover data. This comparison assigns to every fossil sample the average forest cover of its closest modern analogues. This approach, however, regularly underestimates the occurrence of densely forested situations, and should be improved by including marine pollen assemblages in the modern pollen database to better estimate past changes in regional to sub-continental land vegetation cover from deep-sea pollen records. For this reason, we have adopted the simplest approach to infer qualitative changes in forest cover and clearly stated this approach in the revised manuscript. In section “Interglacial climate in South-West Europe during MIS 18 glacial” we have added the following paragraph:

« We qualitatively estimate forest cover changes using arboreal versus non-arboreal pollen percentages. This simple approach, that acknowledges the non-linear relationship between plant abundances and pollen percentages (Zanon et al., 2018), is supported by a wide array of studies on the modern relationship between pollen assemblages and vegetation (e.g., Prentice, 1978 ; Huntley and Birks, 1983) and, specifically, by the recent work of the pollen representation of the vegetation in the Tagus basin ¹⁵ (South-West Iberia), the most important source of pollen preserved at IODP Site U1385¹⁵.”

We have discussed the limitations of our data by adding in section “A warm and wet increasing trend in South-West Europe from MIS 19 to MIS 17” the following paragraph:

« This mismatch between model and proxy reconstructions could be explained by the difficulty in estimating quantitatively the forest cover from pollen data (Zanon et al., 2018). However, our approach identifies substantial qualitative forest cover (winter rainfall) changes that tightly parallel

changes in the T_{therm} SST gradient. Thus, the mismatch could be the result of a feedback process that is not well reproduced in iLOVECLIM such as the poor prediction of the ocean thermal gradient despite good SST simulation.»

Lines 140-141

„Statistically significant correlation coefficient“: this issue with significance is rather tricky with correlations on data that (A) could show temporal serial dependence or (B) non-Gaussian distributions. Dr. Mudelse, amongst others, has done work on that, which needs to be consulted.

We agree with the Reviewer that temporal serial dependence (autocorrelation) and non-Gaussianity are two common properties of paleoclimate time series and this could affect the estimation of the statistical significance of correlation coefficients. For this reason, we have used a sieve bootstrap approach to take into account autocorrelation contained in the paleoclimate time series analyzed in our study when assessing statistical significance for cross-correlation analysis. To carry out this task we have used the R package *funtimes* (Lyubchich et al 2022), which is available on CRAN (<https://cran.r-project.org/package=funtimes>). Figures 4 and 5 show the CCF between the gradient and the MF and the $\delta^{18}\text{O}$ (benthic foraminifera) and the MF, interpolated each 410 years (Figure 4) and 250 years (Figure 5). The correlations are quite similar to the presented in the MS. Thus, these new proposed analyses support our results from a statistical point of view. We have modified Figure 3 in the original manuscript (Figure 4 in the revised manuscript) accordingly.

Figure 4 – Cross-correlation (via Pearson) between the gradient and MF (%) and benthic foraminifera ($\delta^{18}\text{O}$) and MF (%) interpolated each 410 years and using a sieved bootstrap approach to take into account temporal serial dependence (autocorrelation). Black filled circles are the correlation coefficients that are statistically significant. Blue area displays the 95% confidence interval.

Figure 5 - Cross-correlation (via Pearson) between the gradient and MF (%) and benthic foraminifera ($\delta^{18}\text{O}$) and MF (%) interpolated each 250 years and using a sieved bootstrap approach to take into account temporal serial dependence (autocorrelation). Black filled circles are the correlation coefficients that are statistically significant. Blue area displays the 95% confidence interval.

Lines 246-247

The expression „delta18Ob“ is highly unusual in paleoclimate sciences. Perhaps use „delta18O_benth“ (i.e., with subscript) to indicate the benthic character.

Done

Line 250

The unit „cm3“ has to be written with a superscripted „3“.

Done

Line 255

The way to express magnifications is grammatically not correct. One can use, for example, „400- and 1000-fold magnification“.

Done

Line 257

The manuscript states that 4 cm depth resolution corresponds to a time resolution of 450 yr, although the given sedimentation rate (line 243) is 10 cm per year. This is numerically at odds. In line 243 it is written 10 cm/ky, i.e. 10 cm per 1000 years.

Line 260

The manuscript mentions „98 samples out from 126“, while before (line 256) it reports about „One hundred twenty-four“. This is at odds.

This is a mistake. Actually, there are 124 samples. Corrected.

Line 265

While it is OK to mention the software (here: „exactci“) used for the uncertainty calculation, the manuscript must inform readers about the methodological approach plus which statistical

assumptions were made for uncertainty determination (e.g., error propagation assuming Poisson distributions of pollen counts, or bootstrap resampling, etc.).

We have added the following sentence indicating the methodological approach:

“The calculation of CIs is based on the ‘exact’ Clopper–Pearson method assuming a binomial proportion (Scherer, 2018). »

Line 267

The expression „sporo-pollen“ is not correct.

We have replaced with « spores and pollen ».

Line 273

What does the manuscript mean with the wording „robustly identifies“? Robust against which possible violations of made assumptions?

Accordingly to this referee comment, we prefer to delete the term « robustly ».

Line 277

Write „atoms“ instead of „atmos“.

Done

Line 279

The right parenthesis in the expression „(C_37:4)“ has not to be in subscript.

Done

Line 282

Write „2 g“ (i.e., with a space between number and unit).

Done

Line 284

„Mass spectrometer“ has not to be capitalized.

Corrected

Line 285

„Gas chromatograph“ has not to be capitalized.

Corrected

Line 287

„Hydrogen“ is typically not capitalized in scientific texts.

Corrected

Line 289

You cannot put the reference (54) directly after the numerical value (0.015) in order to prevent misinterpretation (0.015 to the power of 54); instead use „0.015 (ref. 54)“.

Done

Line 289

Insert a space after „Uk’_37“.

Done

Line 292

Write „(ref. 57)“.

Done

Line 321

The expression „Oxygen-18-16 isotopes“ is highly unusual.

We replaced the expression with “Oxygen isotopes (¹⁸O and ¹⁶O)”.

Line 331

Write „(ref. 70)“.

Done

Line 332

Write „temperature at 2 m height“.

Done

Line 332-333, the equation

Insert a space after „1000“. Do not write „ln“ in italics. Do not write „H2O“ in italics. Put a comma at the end of the equation.

Done

Lines 335-336, the equation

What does the left subscript „2“ mean? Do not write „calcite VPDB“ in italics. Do not write „SMOW“ in italics. Put a unit to the number (29.98), likely it is „permil“. Put a period at the end of the equation.

We corrected the format of the equation as suggested and added details on the different terms.

Lines 316-338

There is no indication about the temporal resolution of the iLOCECLIM model output.

We specify in the text that the model output are with an annual timestep and that precipitations are also calculated with a monthly timestep.

Line 345

Abbreviate as „J.M.P.-M.“.

Done

Line 432

Write „Scientific Reports“.

Done

Line 473

Reference Stow et al.is not complete.

Corrected

Stow, D. A. V., Hernández-Molina, F. J., Alvarez Zarikian, C. A., and the Expedition 339 Scientists 2013 Proc. IODP, 339: Tokyo (Integrated Ocean Drilling Program Management International, Inc.), doi:10.2204/iodp.proc.339.2013, (2013b).

Line 500

Do not capitalize the title words of the referenced article.

Done

Line 503

Some wrong text („“).

Corrected

Line 505

Reference Goosse incomplete (author list).

Corrected

Line 516

Do not abbreviate journal title.

Corrected

Line 519

The correct journal name is „Journal of the Atmospheric Sciences“.

Corrected

Line 525

Do not abbreviate journal title.

Corrected

Lines 545-550

„LO“ and „TP“ not to be defined since spelled out on figure graphics. Give more details about the „westerlies“: applies for recent and/or paleo?, perhaps stretches much farther north than shown in figure?, and msiing is a reference for paleo westerly position. We have added the requested information « Green arrows: present-day westerlies in a negative mode of the North Atlantic Oscillation (~40°-45°N, Hurrel, 1995), reaching more southern latitudes during the glacial periods as shown by data and model simulations for the Last Glacial Maximum (Lofverstrom, 2020) ».

Figure 1

Give color scale bar (bathymetry, height?). Why are the EAWM arrows left-circling and not right-circling? Place is on northern hemisphere. The abbreviations (e.g., ENACW) are way to difficult to discern: use a uni-color background for those text snippets in the graphics figure.

Done

Line 570-572

„MIS“ is already defined. The colon in „MIS :Marine“ is misplaced. Do not capitalize „Relative“, „Atmospheric“, „Winter“ or „Insolation“.

Corrected

Figure 2

Give units of obliquity. Employ correct ordering of panel labels: from top (a) to bottom (g), not vice versa. „Relative Sea level“: do not capitalize „Sea“. Precession-y-axis to be shifted to the right for better legibility. „[CO₂] ppm“ to be changed into „CO₂ (ppm)“. Subscript index „therm“ too small font size. Panels a, c: define straight lines (e.g., „ordinary least squares straight line fits“). Panel c: define thick and thin lines (likely something with smoothing).

Corrected

Figure 3

Use correct panel labels (i.e., not in boxes). Write unit „ka“ instead of „Ka“. Write „Age (ka)“ instead of „Years BP (Ka)“.

Done

Figure 4

Write unit „ka“ instead of „Ka“.

Corrected

Reviewer #2 (Remarks to the Author):

This paper presents paired changes in SST and pollen-based forest cover from IODP site U1385 spanning the MIS 19-17, aiming to address the potential role of enhanced moisture supply from mid-latitude oceans to higher latitudes in triggering the final onset of significant 100-ka ice cycles. More specifically, the authors propose that MIS 18 is characterized by relatively warm and wet glacial that favors for sustained ice accumulation in the southern Eurasian. Statistical and model results suggest that SST gradient is the likely driver for increasing trends of winter precipitation and forest cover from MIS 19 to 17 on both orbital and millennial timescales. By comparing with East Asian summer monsoon proxies from Chinese loess, the authors conclude that the moisture production in the mid-latitude oceans over the Northern Hemisphere was increased and progressively transported northward during moderate MIS18 glacial, which may result in the full development of the dominant 100-ka ice cycles. The hypothesis is very interesting for better deciphering why the MPT ended around 700 ka. But I have several specific concerns that needs further clarification.

1. During MPT, there are many mild interglacial periods (i.e. MIS23, 27, 33) similar to the climate background of MIS 18 (see Hodell et al., 2022, Clim Past.). The MIS 23 is also followed by a remarkable expansion of the Northern Hemisphere ice during MIS 22. I just wonder whether the authors' hypothesis is also workable in moderate glacial/interglacial periods during the MPT.

Yes, this is an interesting point on which we are currently working on by analysing the pollen content across the entire MPT. Actually, the work by Bahr et al. (2018, Figure 5) also shows a decreasing trend in the T_{therm} SST gradient from MIS 25 to 22 period, similar to that observed from MIS 19 to 17. Therefore, the same processes might be at work explaining the strong ice accumulation during MIS 22. In the last paragraph of the revised main text we have added the following paragraph:

« This process could also be at work in previous moderate glacial/interglacial periods of the MPT. The long-term decreasing T_{therm} SST gradient trend in the North Atlantic from MIS 25 to 22 is similar to that observed from MIS 19 to 17 (Bahr et al., 2018) and could lead to the remarkable expansion of the Northern Hemisphere ice during MIS 22. »

2. Millennial-to-centennial variability. Rapid changes in forest pollen concentration are amazing (Fig. 2a and 4a). Unfortunately, this point is only mentioned in lines 150-155). Not sure whether the mechanisms of coupled winter precipitation and SST gradient on both orbital and millennial timescales are similar not.

The problem here is that the temporal resolution of the T_{therm} SST gradient record is 1000 years between samples and almost three times higher, 400 years, in the pollen percentage record. Therefore, we cannot go further in our interpretation.

3. Simulated rainfall and tree fraction based on iLoveclim model are compared with the proxies from the U1385 (Fig. 2a and 4a). It seems that the amplitudes of model- and pollen-based forest changes are visually different. This amplitude mismatch needs further clarification. If the authors propose that the forest pollen is related to the winter precipitation, I would suggest to add the simulated winter rainfall in Fig. 4a for direct comparison.

We have addressed the mismatch between the amplitude of pollen-based forest cover and simulated tree cover changes in lines 66-71 and lines 147-152 (see above in response to Referee 1). We have simulated and added the winter precipitation in the revised Figs. 3g and 5g.

4. Loess proxies are used to verify intensified summer monsoon and moisture transport during the MIS 18 (Fig. 4 and lines 164-182). These loess proxies are from northwest China (not NE China) and the (Sun et al., 2019, Nat Commun) might be more sensitive to summer monsoon change than the Fe/K ratio. My concern is that in East Asian, enhanced precipitation is occurred in the summer season. How to differentiate the roles of enhanced winter/summer moisture production in mid-latitude North Atlantic/northwest Pacific in accumulating the ice expansion at the end of the MPT.

We have replaced NE China with North China.

Based on the study of the present-day $\delta^{13}\text{C}_c$ -climate relationship from 20 surface soil samples over the Chinese Loess Plateau, Sun et al. (2019, Nat. Comm.) have found that the correlation between the $\delta^{13}\text{C}_c$ values and mean annual/summer precipitation is indeed quite similar. The same lead author states in his Nat. Geosci. paper (2021) that the Fe/K ratio is also a proxy for East Asian summer monsoon. Therefore, we take both proxies as indicators of EASM without evaluating if one is better than the other.

We think that both, winter North Atlantic and late summer West North Pacific moisture production have fed the ice caps across the MPT. However, a modeling approach should be applied to differentiate the roles of each of them.

The authors suggest that the oceanic moisture production plays a key role in explaining the strong winter precipitation during a glacial period (lines 202-206). I wonder whether iLOVECLIM simulation results can demonstrate this enhanced moisture transport during MIS 18 relative to MIS 19. The differences in moisture transport flux and precipitation between MIS 18 and 19 can be inferred from the transient modeling results.

Annual and seasonal precipitation for MIS 18 and 19 has been simulated with iLOVECLIM and the simulated values are included in the new figures. These simulations show that during MIS 18 winter precipitation is higher compared to MIS 19. iLOVECLIM is, in contrast, unable to simulate moisture transport flux. We have, however, simulated the relative humidity between sites U1385 and U1313 in the North Atlantic and between ODP Sites 882 and 1143 in the West North Pacific across the studied period. The simulations show the northward migration of the moisture source from MIS 19 to 18 in both regions, in agreement with data. These new results are presented in Figure S7 of the Supplementary information and included in the revised manuscript. These new results have been added in the revised manuscript (lines 180-184 and 218-219).

Reviewer #3 (Remarks to the Author):

Dear authors,

I think that your paper titled “Interglacial climates during the last glacial of the Middle Pleistocene Transition” is a good one and that the data you use (even if already partly published) allow new advances in the knowledge of the MPT climate.

I have one main concern:

- How can you state that the expansions of deciduous oaks are all components of the Mediterranean forest? This idea risks to be an exaggeration of the principle of actualism. We well know that during the Mid Pleistocene Transition the vegetation was quite different from the present one, with the presence of many arboreal species growing in wet environments, that are at present extinct in Europe. Deciduous oaks pollen could therefore include also oaks living in wet environments and not only in relative dry conditions. This wider interpretation would fit more with the Balkan terrestrial records (Tenaghi Philippon and Ohrid) in which deciduous oaks are included in the mesophilous vegetation.

For clarifying this issue we have added the following paragraph in section « Interglacial climate in South-West Europe during MIS 18 glacial » (lines 95-106):

« At Site U1385 the expansion of the deciduous oaks parallels the expansion of the Mediterranean sclerophyllous taxa such as evergreen *Quercus*, *Olea*, *Phyllirea*, *Pistacia*... The peaks of deciduous oak pollen percentages (~40-60 %) correspond with the peaks in the pollen percentages of the sclerophyllous elements (~10-20 %). The pollen assemblages recorded at this site during the warm phases are, therefore, very similar to the modern ones inferred from samples collected in the deciduous oak woods of the Tagus basin characterized by ~10-20 % of sclerophyllous pollen taxa from the thermomediterranean belt and ~40-60 % of deciduous oak pollen taxa from mesomediterranean belt (Morales-Molino et al., 2020). No arboreal trees extinct at present in Europe are recorded during this interval at Site U1385. The vegetation inferred from the pollen assemblages at site U1385 is, therefore, very close to the present-day Mediterranean forest of SW Iberia, composed of broadleaf trees, such as the oak and mixed sclerophyll forests (Polunin and Walters, 1985).”

I include the original files with many comments and minor issues, please consider them. There are also minor issues in the figures (e.g. red and green used in the same curve) that should be adjusted.

Best regards

Laura Sadori

Comments on the original files of the manuscript

Line 108 - please include a column with MIS numbers

We do not understand the request. The MIS numbers are included in Figure 2 already.

Line 110 - why? summer aridity is mostly preventing the expansion of deciduous vegetation, this is the main constrain. With the same winter (high) precipitation we can have either mesophilous and mediterranean vegetation, prolonged summer aridity is the limiting factor for deciduous vegetation and therefore, the favoring factor for sclerophyllous mediterranean vegetation. The Mediterranean woody flora is strongly dominated by stress tolerance-related traits.

I realized that the main problem is that you include in Mediterranean forest (Fig. S1) deciduous oaks, that at the terrestrial sites of Ohrid and TP are included in mesophyllous vegetation, so the comparison among pollen records using "groups" became impossible.

As stated in the manuscript, our interpretation of changes in Mediterranean forest cover and winter precipitations is based on: a) the strong statistical correlation between the NAO index, controlling the intensity and direction of the westerlies, i.e. winter precipitation, and the present-day Mediterranean forest cover changes (including thermomediterranean and mesomediterranean floristic elements), inferred from the Mediterranean forest pollen percentages, in SW Iberia (Gouveia et al., 2008), and b) transient model simulations with time-varying insolation and atmospheric CO₂ concentrations showing that winter precipitation is the main factor controlling Mediterranean forest cover changes during past interglacials and the Holocene (Oliveira et al., 2018).

Lines 128-129 - yes, it sounds possible. The best way is to look at available long pollen terrestrial record. The closest ones are from the Balkan peninsula, e.g. Tenaghi Philippon and Ohrid.

This comparison is presented at the end of section "A warm and wet increasing trend in South-West Europe from MIS 19 to MIS 17" (lines 222-226 of the revised manuscript).

Lines 131-132 – "again no clear to me" the sentence more rainfall during MIS 18 glacial compared to MIS 19 interglacial.

This interpretation is based on: a) present-day observations showing that winter precipitation is the main factor controlling the Mediterranean forest development in SW Iberia (Gouveia et al., 2008), and b) the combination of model simulations and data for past interglacials showing that Mediterranean forest pollen percentage changes, qualitative changes in forest cover, are mainly the result of changes in winter precipitation as well (Oliveira et al., 2018). Moreover, the growth of evergreen oaks, the second most important component of the Mediterranean forest, also requires winter precipitation as summers are dry. Therefore, the higher pollen percentages of the Mediterranean forest during MIS 18 compared to MIS 19 indicate more winter rainfall during MIS 18 than during MIS 19.

Line 146 - the trend you mention is not so clear, and, again, to relate winter precipitation to the expansion of mediterranean vegetation needs a physiological explanation not known in literature.

Please see comments above.

Line 156 – 730 ?

We have replaced with 725 ka.

Line 159 - Ohrid is a transboundary lake, the drilling was indeed performed in North Macedonia
For Ohrid have also a look at Donders et al., 2022, PNAS.

We have replaced Albania with « Albania/North Macedonia » and cited in the appropriate Figure legend (Fig. S7) the reference to Donders et al. (2022).

Line 160 - which are the data from central Europe?

Thanks, it was a mistake. We have deleted Central Europe.

Line 161 - why should we expect a linear forest increase? I am not sure of it looking at fig. 2a. Moreover, please consider that mediterranean vegetation (as you include deciduous oaks in it), very well represented in U1385, is quite rare at Ohrid (see Donders et al., PNAS, 2021).

We do not expect the same linear forest increase in SE Europe and we have given a tentative explanation for that.

Lines 166-167 - during MIS18 this is not true, we can say that there is an increase in rainfall in MIS17 in comparison to MIS 19, even if increased precipitations are found during the forest expansions of MIS18

As far as the Mediterranean forest pollen percentages are concerned, we infer higher winter rainfall during MIS 18 (~60% of Mediterranean forest pollen) than during MIS 19 (~40% of Mediterranean forest pollen).

We have completed reference 9:

Ehlers, J., & Gibbard, P.L. 2007 Glaciation: overview. In: Elias, S.A. (ed.) Encyclopedia of Quaternary Science. Elsevier: Amsterdam. 1023-1031.

Comments on the Additional information file

Figure S1 - please pay attention: the use of red together with green is not proper. Daltonic people will be not able to distinguish the two colors.

Done. We have replaced red with yellow.

Figure S2 - the two tonalities of green are not easily to distinguish. I wonder if a shaded area could be a solution.

Done

Figure S3 - Again, red and green. The colors themselves provide an indication of aridity and sclerophilly, I agree, but they cannot be used together. It is important to know which pollen types have been included as Mediterranean forest includes sclerophyllous.

We moved this figure to the main text. This figure, Figure 2 in the revised version of the manuscript, is important to illustrate more in detail the revised text in section « Interglacial climate in South-West Europe during MIS 18 glacial ». We have replaced red with yellow and added the pollen types included in the sclerophyllous.

Figure S5 - Are all citations present?

No references to cite. This graph includes the simulations performed in this study.

Figure S8 - which is which? citations of original papers are missing. Which plants have been included in the two curves?

We have indicated the references for the Tenaghi Philippon and Lake Ohrid pollen sequences and added the floristic components of the Temperate forest pollen percentage curves in the two sites.

Reviewer #1 (Remarks to the Author):

Sanchez-Goni-et al-2023-Nat-Comm-Revision-01

Review

The authors did a good job on the first revision, but not a perfect one. Below I go briefly through the major points [1-2] raised in my initial review and also through the left open minor points.

[1] Proxy knowledge, previous work

Adequate response, new literature cited: now I believe that the proxy quality is OK and that previous work is properly cited.

[2] Correlation, binned rXY vs WLMC

Not adequately dealt with. Explanation:

(a) You do not need a "multivariate technique", you are on one hand rather interested in a time-dependence of the strength of association. In that regard, wavelet methods may be adequate. The relevant methodical paper on WLMC is Fernandez-Macho (2018) *Physica A*. That is certainly an interesting paper, but it has a deficit because the new WLMC method developed therein is not properly backed up by means of a simulation study (ie., many thousand runs, such that uncertainty measures for WLMC can be reliably assessed) --- it only has two synthetic examples (each with one run only). WLMC has the methodological deficit that it requires interpolation to equidistance, which means a step away from the original data, and it may lead to spurious effects (also within wavelet spectra). Your statement from the response that "The number of elements of the new regular (binned) time series is very small" and "correlations estimated are not reliable" is contradicted by an own analysis of your data (thanks for making them available); more details below (c).

(b) You basically need on the other hand a correlation estimation to test the association between pollen and SST_therm_gradient, which you currently do via CCF. However, the rXY method achieves the same task (c), and in my view more reliably.

(c) I did a reanalysis of your data (pollen, SST_therm_gradient) using binning (rXY), which is contained as a separate file (analysis_rxy.pdf). This analysis shows that the estimated binned correlation coefficient is $r_{XY} = -0.568$ with a 95% confidence interval of [-0.751; -0.303] --- that is, it confirms your assessment.

(d) Summarizing the correlation aspect in the manuscript: indeed, an rXY analysis can shed more light on the issue, as was already indicated in my first review. It is assessed as mandatory to include the r_{XY} analysis (numerical value and citations in main text, plots in supplement). *Therefore, I should like to see the next revision.*

Minor points (line numbers refer to my initial review of the initial manuscript)

Line 1

Title phrase still sounds ugly to me. How can an "interglacial climate" be within an "ultimate glacial"? This could alienate paleoclimate readers. Use the word "stage" or "phase" or similar to indicate that you observe *within an existing glacial* some milder phases or stages or conditions.

Lines 26-27

Cf. Line 1.

Line 93

What I meant with my recommendation to not put reference numbers behind (as superscripts) of "real numbers" or numerical expressions in the manuscript; here you write "U1385" with a superscripted "3", and a reader may wonder whether 1385 has to be raised to the power of 3.

Lines 140-141 and Figure 4 of response.

The "blu" (sic!) area certainly does not display a 95% CI for the correlation coefficients (black filled circles) - please note that for many time lags, the estimate is clearly outside of the CI - which does not make any sense! It rather seems that you calculated upper and lower bounds of the correlation estimation under a null hypothesis of zero

correlation – which is not the same as a CI.

Line 265

The new reference given by you (Scherer, 2018) is just a reference to an R package, not a properly reviewed statistical methodological research article. You must give an additional source.

Line 279

Contrary to what you claim in your response, that error has not been corrected!

Lines 500, 503, 505, 516, 519, 525

Contrary to what you claim in your response, these errors have not been corrected!

Figure 1 (map)

My question (EAWM arrows leftwise circling) has not been dealt with (neither in response, nor in revision).

Best wishes, Manfred Mudelsee

Reviewer #1 Attachment on the following page.

data (tx, x): [path + filename]

Pollen.txt

data (ty, y): [path + filename]

SSTgradient.txt

Data size (x series): nx = 310

Data size (y series): ny = 127

Start: tx(1) = 668.489

End: tx(nx) = 795.200

Average spacing (tx, x): 0.410

Start: ty(1) = 670.000

End: ty(ny) = 796.000

Average spacing (ty, y): 1.000

Bin width -- minimum: 0.292

-- maximum: 127.511

Bin width -- INPUT

1.153

PearsonT3 (Version 3.0.2, September 2022)

data (t, x, y) [path + filename]

Binning-data--Pollen-SSTgradient.dat

Number of threads: 24

1st bootstrap loop finished

2nd bootstrap loop working ...

2nd bootstrap loop working ... 500 / 2000

2nd bootstrap loop working ... 1000 / 2000

2nd bootstrap loop working ... 1500 / 2000

2nd bootstrap loop working ... 2000 / 2000

2nd bootstrap loop finished

Data file name: Binning-data--Pollen-SSTgradient.dat

Time interval - original: [670.218; 794.742] - 109 points

Persistence time (x): 6.727

Persistence time (y): 3.553

Pairwise-MBB block length: 14

Confidence interval type: Calibrated Student's t

Pearson's r, 95 % confidence interval: -0.568 [-0.751; -0.303]

Reviewer #2 (Remarks to the Author):

My major concern about proxy-model comparison has been well addressed in the revision. I'm satisfied with the response and have no further suggestions.

Reviewer #3 (Remarks to the Author):

Dear authors,

I found the manuscript pretty improved and I fully respect your opinions as they are based on independent data.

As far as I am concerned I am OK with this new version.

Best regards

We thank Reviewer 1 for his careful re-reading of the manuscript and highlighting the remaining mistakes.

Reviewer #1 (Remarks to the Author):

Sanchez-Goni-et-al-2023-Nat-Comm-Revision-01

Review

The authors did a good job on the first revision, but not a perfect one. Below I go briefly through the major points [1-2] raised in my initial review and also through the left open minor points.

[1] Proxy knowledge, previous work

Adequate response, new literature cited: now I believe that the proxy quality is OK and that previous work is properly cited.

[2] Correlation, binned rXY vs WLMC

Not adequately dealt with. Explanation:

(a) You do not need a “multivariate technique”, you are on one hand rather interested in a time-dependence of the strength of association. In that regard, wavelet methods may be adequate. The relevant methodical paper on WLMC is Fernandez-Macho (2018) Physica A. That is certainly an interesting paper, but it has a deficit because the new WLMC method developed therein is not properly backed up by means of a simulation study (ie., many thousand runs, such that uncertainty measures for WLMC can be reliably assessed) --- it only has two synthetic examples (each with one run only). WLMC has the methodological deficit that it requires interpolation to equidistance, which means a step away from the original data, and it may lead to spurious effects (also within wavelet spectra). Your statement from the response that “The number of elements of the new regular (binned) time series is very small” and “correlations estimated are not reliable” is contradicted by an own analysis of your data (thanks for making them available); more details below (c).

(b) You basically need on the other hand a correlation estimation to test the association between pollen and SST_therm_gradient, which you currently do via CCF. However, the rXY method achieves the same task (c), and in my view more reliably.

(c) I did a reanalysis of your data (pollen, SST_therm_gradient) using binning (rXY), which is contained as a separate file (analysis_rxy.pdf). This analysis shows that the estimated binned correlation coefficient is $r_{XY} = -0.568$ with a 95% confidence interval of $[-0.751; -0.303]$ --- that is, it confirms your assessment.

(d) Summarizing the correlation aspect in the manuscript: indeed, an rXY analysis can shed more light on the issue, as was already indicated in my first review. It is assessed as mandatory to include the r_{XY} analysis (numerical value and citations in main text, plots in supplement). *Therefore, I should like to see the next revision.*

We wish to express our gratitude to Reviewer number 1, Dr. Manfred Mudelsee, for his useful comments and suggestions, especially the suggestion for the statistical data analysis techniques. The application of these techniques to our data has substantially improved the quality of our MS. Following the recommendation of Dr. Mudelsee, we have replaced in our analysis the wavelet local multiple correlation (WLMC) and cross correlation (CCF) analyses with the binned correlation (Mudelsee 2010, 2014) and the Pearson correlation method (2003). For this reason, Figure 4 and Figures S4 and S5 have been removed and we have added a new Figure S4 to the Supplementary Information.

Minor points (line numbers refer to my initial review of the initial manuscript)

Line 1

Title phrase still sounds ugly to me. How can an “interglacial climate” be within an “ultimate glacial”? This could alienate paleoclimate readers. Use the word “stage” or “phase” or similar to indicate that you observe *within an existing glacial* some milder phases or stages or conditions.

We have replaced the title with the following one : « Moist and warm conditions in Eurasia during the last glacial of the Middle Pleistocene Transition »

Lines 26-27

Cf. Line 1.

We have replaced « interglacial » with « high »

Line 93

What I meant with my recommendation to not put reference numbers behind (as superscripts) of “real numbers” or numerical expressions in the manuscript; here you write “U1385” with a superscripted “3”, and a reader may wonder whether 1385 has to be raised to the power of 3.

We have replaced « U1385³ » with « U1385 (ref. 3) »

Lines 140-141 and Figure 4 of response.

The “blu” (sic!) area certainly does not display a 95% CI for the correlation coefficients (black filled circles) – please note that for many time lags, the estimate is clearly outside of the CI – which does not make any sense! It rather seems that you calculated upper and lower bounds of the correlation estimation under a null hypothesis of zero correlation – which is not the same as a CI.

We agree with the Reviewer. After applying the new and robust statistical methods recommended by him, we have decided to remove Figures 4a and 4b, and just added the Pearson's correlation coefficients + CI in the MS. A visual comparison between original and binned time series are shown in the new Figure S4.

Line 265

The new reference given by you (Scherer, 2018) is just a reference to an R package, not a properly reviewed statistical methodological research article. You must give an additional source.

Thank you for the comment. The method is the so-called Clopper-Pearson exact CI and the reference is : Clopper, C. and Pearson, E.S. (1934) The use of confidence or fiducial limits illustrated in the case of the binomial. *Biometrika* 26, 404-413.

Line 279

Contrary to what you claim in your response, that error has not been corrected!

We have corrected. Now it reads « (C_{37:4}) »

Lines 500, 503, 505, 516, 519, 525

Contrary to what you claim in your response, these errors have not been corrected!

We have completed the references in lines 500, 503, 516, 519, 525 (new lines 602, 616, 627, 630, 636). However, reference in line 505 seems to us correct as the format of references in Nature journals replace with « et al. » when the number of authors is more than 5.

Figure 1 (map)

My question (EAWM arrows leftwise circling) has not been dealt with (neither in response, nor in revision).

We have corrected the direction of the arrows indicating the EASM, and added the reference of Porter and An (1995) to support it.

Reviewer #2 (Remarks to the Author):

My major concern about proxy-model comparison has been well addressed in the revision. I'm satisfied with the response and have no further suggestions.

Reviewer #3 (Remarks to the Author):

Dear authors,

I found the manuscript pretty improved and I fully respect your opinions as they are based on independent data.

As far as I am concerned I am OK with this new version.

Best regards

Reviewer #1 (Remarks to the Author):

The second revision further improved the manuscript substantially. Judged from my side, it is now fit for publication in Nature Communications!

Congratulations to the science team for doing that excellent job and to the Nature Communications editors for securing another great paper!

Best wishes, Manfred Mudelsee